# Contour Mission Flight Planning of UAV for Photogrammetric in Hillside Areas

Chia-Sheng Hsieh , Darn-Horng Hsiao * and Di-Yi Lin

Department of Civil Engineering, National Kaohsiung University of Science and Technology, 415 Chien Kung Road, Kaohsiung 80778, Taiwan; hsieh@nkust.edu.tw (C.-S.H.); daniellinboy1@gmail.com (D.-Y.L.)
* Correspondence: hsiaodh@nkust.edu.tw

**Abstract:** Unmanned Aerial Vehicle (UAV) photogrammetry is an effective method for acquiring terrain information. However, in hillside areas, the terrain is complex, and the altitude varies greatly. The mission flight is planned by using equal altitude; in the actual shooting, the geometry and resolution of the pixel within the same image or between adjacent images will be inconsistent due to the different shooting distances. The number and accuracy of point clouds are affected. We propose a contour mission flight plan method, which involves designing flight plans based on the existing digital elevation model (DEM) and the desired flight altitude. This method for aerial photography is more effective in maintaining a consistent ground shooting distance during image capture. Experiments were conducted using a simulated DEM and the undulating terrain of the Kaohsiung Liugui area in Taiwan to verify the effect of contour mission flight planning in the hillside area. The results show that, due to the significant variation of terrain in the hillside area, the use of a contour mission flight plan for aerial photography can be more consistent with the originally planned altitude but requires more planning and operating time. The minor height difference, higher overlap, and improved accuracy of the results show that contour mission planning can provide a suitable solution for UAVs in hillside areas.

**Keywords:** UAV; mission planning; hillside; photogrammetry

## 1. Introduction

The development of remote sensing platforms and detailed topographic surveys is the key to and a prerequisite for many studies in Earth science [1,2]. The recent and continuous development of unmanned aerial vehicles (UAVs) promotes new remote sensing platforms that can supply ultra-high spatial and temporal resolution [3,4]. The small UAVs continue to gain importance in remote sensing applications in Earth science. Thus, one of the essential advantages of UAVs is that they can easily capture video or overlapping images in permitted areas and under suitable atmospheric conditions [5].

With a consumer camera, UAVs can obtain high-resolution optical images, leading to increased scientific studies on UAV-based remote sensing of landslides [6–10] and glacier variations [11–14]. The combination of UAV-based aerial images and Structure from Motion (SfM) technology provides a rapid and cost-effective composition for remote sensing and monitoring landslides and glaciers. SfM is a photogrammetric method for creating 3D models of a feature or topography from overlapping 2D photos taken from many locations and orientations to reconstruct the photographed scene [15]. The combination of these two technologies can provide practical value. However, since SfM photogrammetry depends on the input image data quality, the UAV images' quality must be carefully considered.

UAVs can be flown either manually or using autopilot. Manual flights depend on the operator's experience and skill and can result in low overlap and large rotation angles, affecting the 3D model reconstruction [16,17]. Most researchers prefer autopilot flights based on route planning for capturing [18,19]. However, flight missions must consider

the distance variation to the object, which can cause problems in stereoscopic coverage or result in significant differences in ground sampling distance (GSD) [20–22]. The GSD is the relationship between camera equipment parameters (camera focal length, camera pixel) and flight altitude above ground level (AGL) [23]. It is essential to evaluate GSD size before capturing UAV images. As a result, UAV-based topographic reconstructions of mountainous environments are often considered less accurate than those carried out in other flat terrain settings.

In studies related to flight planning in mountainous areas, many researchers have proposed solutions for some characteristics of mountainous terrain variations. During the mission, it is desirable to maintain a constant flight altitude relative to the ground surface to obtain a uniform ground resolution. In a study of volcano monitoring, De Beni et al. consider that the altitude of the UAV mission is not absolute but is always referenced to the altitude of the origin (take-off point). When the whole survey area is decomposed into sub-regions, special attention needs to be paid to referring the altitude of each mission to a specific origin [24]. Manconi et al. also consider the existing topography of the survey area. In areas with steep slopes, very high-resolution digital models and orthophotos are needed to analyze the geometry of the fracture surfaces in detail. The terrain is divided into blocks according to its elevation. Then the flight mission is carried out separately according to the blocks to maintain the same shooting distance to the ground [25].

Agüera-Vega et al. dealt with extreme topography in an almost vertical road cut-slope in Spain using a UAV. They took horizontal images in four flight lines on a vertical plane, and oblique images were taken at 45 degrees downwards in two passes [26]. Valkaniotis et al. mapped landslides associated with earthquakes in Greece. The actual UAV survey covered the landslide area with vertical and oblique views [27]. Trajkovski et al. proposed a method that combines vertical and oblique images to capture steep terrain [28]. Gomez-Lopez suggests that schemas are the combined classical block flight and corridor flight in one mission for slope and line objects [19]. The methods described above have improved the suitability of UAV surveys in steep terrain by tilting the photo or defining different heights of the flight routes.

In this study, we adopt the concept of contour to plan the UAV flight mission. Free digital elevation model (DEM) data can easily generate the contour. We utilize the contour information to adjust the position and height of the flight route to follow the terrain while maintaining a consistent AGL height and ensuring sufficient image overlap. The primary aim is to reduce the variation in elevation of each image, as this can cause mismatches in image features.

## 2. Methodology

This study describes the methods used for flight planning in mountainous areas. Two mission flight planning methods were used: the grid-based mission commonly used in UAV surveys and the contour-based mission proposed in this study. The following sections describe each process, outlining their respective procedures and considerations.

### 2.1. Grid-Type Flight Plan Method

The grid-type mission is generally adopted in mountain terrain. The route is planned according to DEM calculations of flight level, which consider the average terrain elevation of the corresponding route. The analysis of flight height is based on the corresponding average elevation of the route position plus the shooting distance. The UAV collects images at the same height level and then rises to the next route height to capture images until the planned route is completed, as shown in Figure 1.

When the UAV uses the same altitude to shoot vertically, if the area is flat, the UAV can keep the same altitude to take photos. However, when photographing in mountainous regions, if you still choose to keep the same altitude for photographs, it will affect the image shooting distance, resulting in an impact on the GSD and overlap, as shown in Figure 2. Figure 2a shows that when UAVs photograph at the same altitude, the shooting

distance is shortened as the terrain rises (a > b), resulting in a significant difference in image resolution, and the ground image area that can be covered becomes smaller (c > d > e), as shown in Figure 2b, which reduces the image overlap. The above scenario shows that aerial photography at the same altitude will result in inconsistent image quality when taking pictures of rugged slope areas.

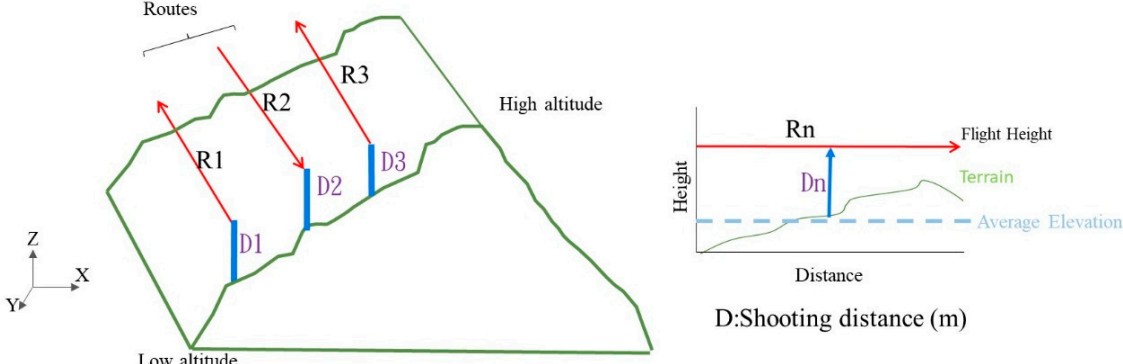

**Figure 1.** The route planning and actual image location site of single grid-type. The Rn is the nth route, and Dn is the shooting distance of the nth route.

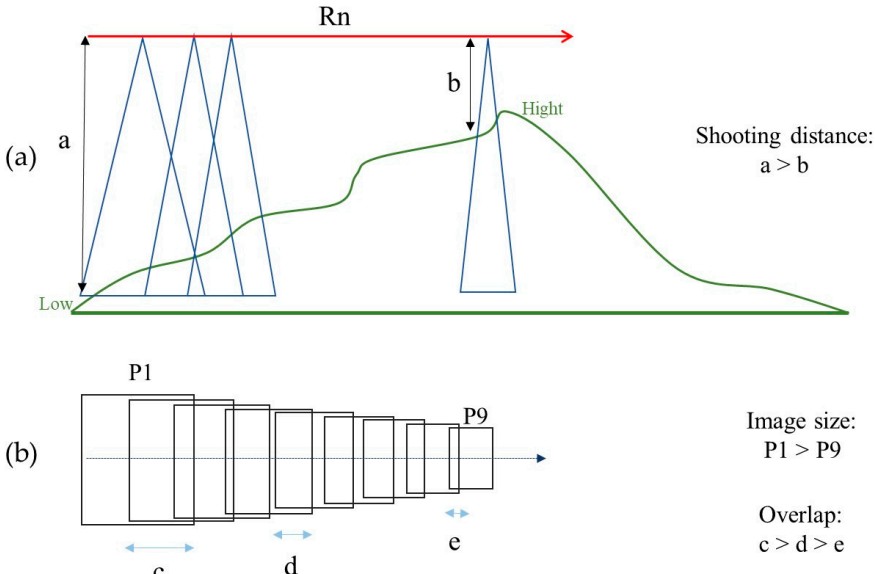

**Figure 2.** The schematic scene of the common problem for UAV capture photos in the rugged slope area. The transverse flight Rn for the slope is similar Rn in Figure 1. (**a**) The figure shows the change in shooting distance due to the difference in ground height when using same altitude photography in the slope area. (**b**) The image size and overlap differ due to different shooting distances. Where a, b are the different shooting distances, Pn is the size of the nth image, and c, d, e are the various overlaps.

### 2.2. Contour Mission Flight Planning

This study proposes a method to keep the shooting distance consistent based on the contour principle and DEM information to maintain the image quality as much as possible.

The method proposed in this study to generate contour-type aerial routes is shown in the flowchart in Figure 3, steps S1–S7. First, in S1, the user provides information about the aerial photo area, camera, and mapping requirements. In S2, the system extracts the terrain information from the DEM and calculates the flying height and shooting baseline. In S3, the location of the first flight line is calculated based on the relationship between

the lowest and highest elevations and the boundary line. In S4, the system will consider the distance between the routes and the topographic changes corresponding to the routes, and use either S5 or S6 to generate the routes, depending on the situation, until the highest point elevation is calculated and the routes for the whole area are planned. The entire process uses the principle of contour to maintain the consistency of the shooting distance, which can improve the image quality and coverage, thus improving the effect of aerial photography by UAV.

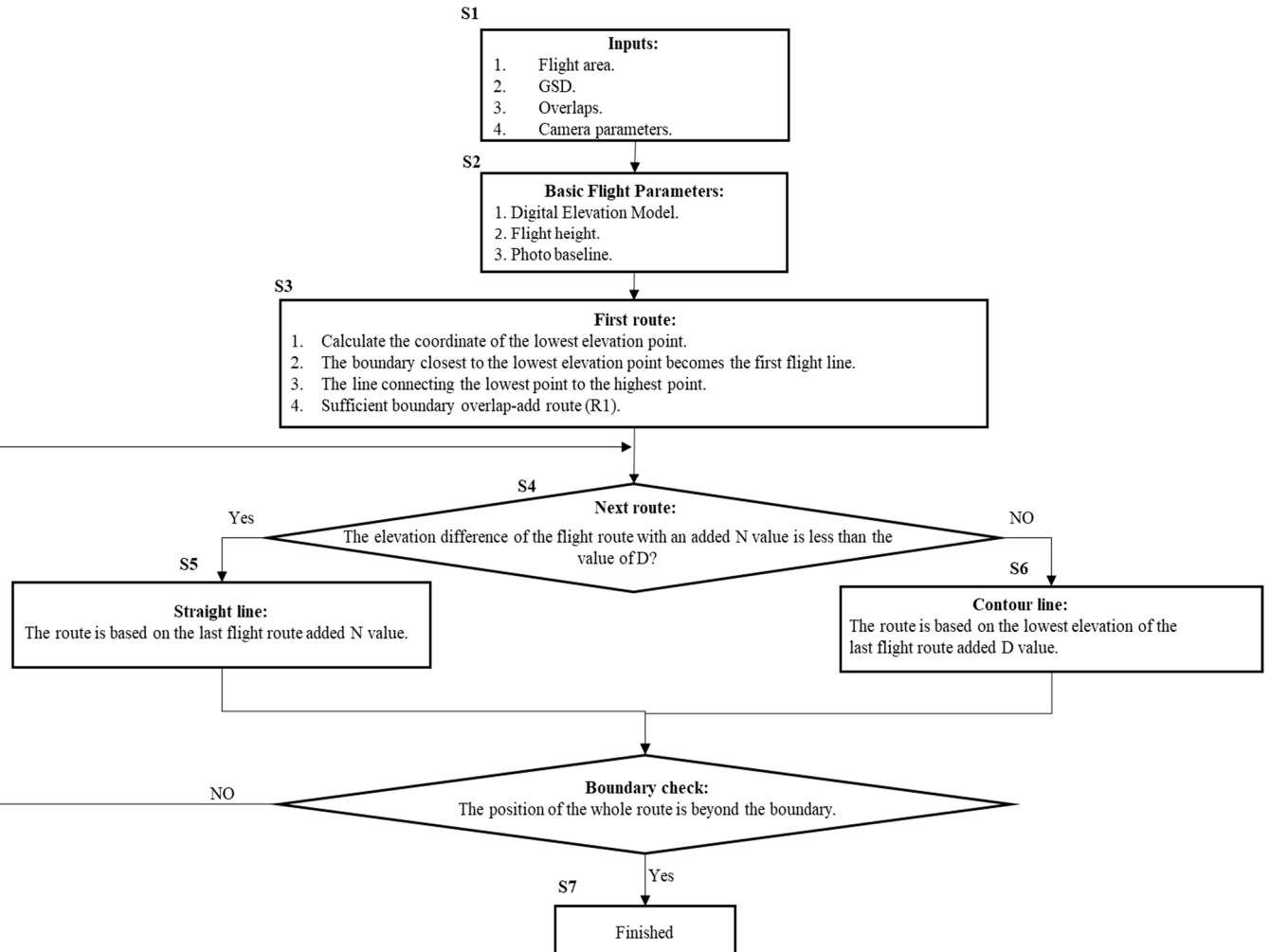

**Figure 3.** Contour route planning flow chart.

The following are specific descriptions of the various steps in the process.

### 2.2.1. S1: User-Provided Parameters

- Flight area: set the boundary coordinates of the working area.
- GSD: ground sample distance (m/pixel).
- Camera parameters: focal length, image size, sensor size.
- Overlap and sidelap:
- UAV images require a high percentage overlap to generate high-quality 3D point clouds using the overlay information from the photogrammetric process. There is a positive relationship between the image overlap and the accuracy of digital elevation models (DEMs). The accuracy is increased with the increased overlap percentage, and the object's shape is optimized. Photogrammetric software, such as Agisoft Metashape [29], recommends that UAV images be acquired with almost 80% overlap

and 60% sidelap. Pix4DMapper 4.1 suggests at least 75% overlap and 60% sidelap [30]; generally, the overlap is equal to or greater than the sidelap.

### 2.2.2. S2: Calculate the Basic Flight Parameters

- Import of flight area DEM: DEMs with a particular spatial resolution can be provided worldwide (mostly in low resolution), nationwide (mostly in different resolutions), and locally from previous surveys. Worldwide, the DEM has Shuttle Radar Topographic Mission (SRTM) and Advanced Spaceborne Thermal Emission and Reflection Radiometer (ASTER). Due to the SRTM spatial resolution of 3 arc seconds [31], the SRTM DEM must be more accurate for planning a UAV mission in steep hillside terrain [32]. The ASTER (1 arc second) DEM was selected for the study. The ASTER DEM is used to access publicly available data repositories such as the NASA Earth Observing System Data. Users can download the desired DEM files from the platform in the appropriate flight area.
- Flight height: The user provides the aerial photo area, as well as the conditions for capturing images (such as GSD and overlap), the camera parameters (focal length, sensor size, image resolution), according to the GSD and sensor parameters can calculate the AGL, as Equation (1).

$$AGL = \frac{f \times GSD \times W}{w} \tag{1}$$

where *AGL* is the flight altitude above ground level (m), $f$ is the focal length (mm), *GSD* is the ground sample distance (m/pixel), *W* is the image width (pixel), and $w$ is the sensor width (mm).

Most topographic surveys have a scale of 1/300–1/500, and the corresponding *GSD* is 3–5 cm. Usually, the 3–5 cm *GSD* flight altitude is 150–250 m. The lower the flight altitude, the higher the *GSD*, but the smaller the image coverage, the more images will be taken, and additional flight and image post-processing time will be required.

$$B = GSD \times W(1 - p) \tag{2}$$

where *B* is the photo baseline (m), and $p$ is overlap (%).

### 2.2.3. S3: Calculating First Flight Route

The system calculates the lowest elevation position in the working area and uses the nearest boundary of the lowest position as the location where the data must be obtained. Then, the system will generate the first route (R1) from the border to the lower slope. The first route is generated by linking the lowest and highest elevations to obtain the direction of the lower slope. The single grid method generates the first route to the lower slope to ensure that the topographic information at the boundary can be obtained.

### 2.2.4. S4: Next Route

When calculating the next route, two parameters are considered, the adjacent routes spacing *N* and the elevation difference value *D*. First, the system evaluates the change in elevation of the corresponding terrain based on the position of the previous route plus the *N* value. The system then determines the relationship between the elevation change and the elevation difference threshold *D* and decides whether to produce the next route in step S5 or S6. This approach ensures that the route is reasonable and the shooting distance is accurate. The decision N value is based on estimated image accuracy [33]. N value is the spacing between adjacent flight routes (m), as shown in Equation (3).

$$N = GSD \times H(1 - q) \tag{3}$$

where *H* is image height (pixel), and $q$ is sidelap (%).

The *D* value is a threshold value of difference elevation (m). It is based on estimated image accuracy [33], as shown in Equation (4).

$$D = \sqrt{\frac{GSD \times B \times f}{\sigma_i}} \tag{4}$$

where *D* is the threshold value of difference elevation, and $\sigma_i$ is image measurement accuracy.

### 2.2.5. S5: Straight Route

The route is a straight line when the *N* value is added to the route, and the corresponding elevation undulation is lower than the *D* value. This means that the difference in terrain between the two routes is slight, and the following route can be produced directly using the single grid.

### 2.2.6. S6: Contour Route

The route is a contour line when the *N* value is added to the route, and the corresponding elevation is higher than the *D* value. The contour value is the elevation of the lowest point of the previous route plus the *D* value.

### 2.2.7. S7: Boundary Check

Determine if the entire route is located outside the boundary. If not, proceed to the next route. If yes, end the whole step.

Through the contour route planning process S1–S7 proposed by the flowchart, the route paths of the whole region can be successfully obtained. This process uses the user-supplied parameters and DEM information to plan a suitable aerial route considering the topographic relief. Our method allows each route to maintain the same flight altitude, resulting in a more continuous GSD and avoiding the UAV's constant vertical displacement. This method is more effective in keeping the smoothness of the route than allowing each aerial point to build its altitude. Through such a process, we can effectively consider the terrain up and down and generate a suitable aerial path, providing a better solution for UAV aerial photography missions.

This study utilized the ASTER DEM as the terrain reference data, with the coordinate system set to WGS-84. The DEM was extracted based on the user-defined flight area. The flight route was then generated using the defined camera parameters, ground sample distance (GSD), and overlap rate, following the workflow outlined in Figure 3, steps S1–S7. The calculations were performed using Python code, resulting in a KML file format. The IPad's DJI Ground Station Pro app was used to import the KML file and create the flight mission, allowing flight speed selection. Subsequently, the UAV executed the flight mission to capture aerial images.

## 3. Simulation Test

To verify the differential effectiveness of the contour-type flight planning method proposed in this study in mountainous areas, we will discuss it through two simulation zones. Simulation Case 1 is a general slope in Nanhua District, and Simulation Case 2 is a hillside terrain in Maolin District where landslides occur. Two regions with different surface changes were selected. The existing DEMs were used to simulate the planning and to compare and analyze the differences. General users commonly use a GSD of 4 cm and an overlap rate of 80% for planning. The calculated flying height is 200 m, the flight route spacing N is 140 m, and the elevation difference threshold (D) is 34 m.

### 3.1. Case 1: Nanhua District

Case 1 is located in Nanhua District, with an elevation of 190–300 m, a slope of 24 degrees, and an area of 60,000 m$^2$. This study used a grid and contour simulation for

route planning, as shown in Figure 4. The grid-type routes were divided into 5 routes with 35 images, while the contour-type routes were divided into 14 routes with 187 images.

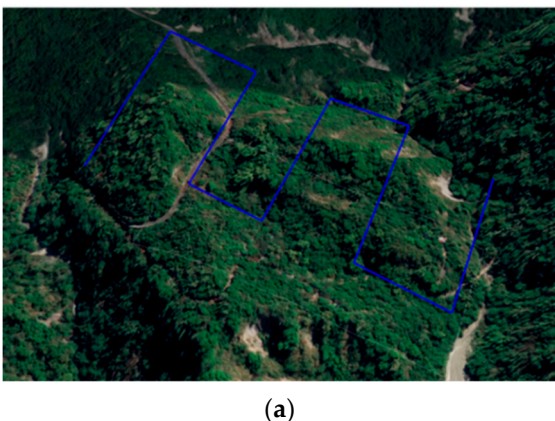

(**a**)

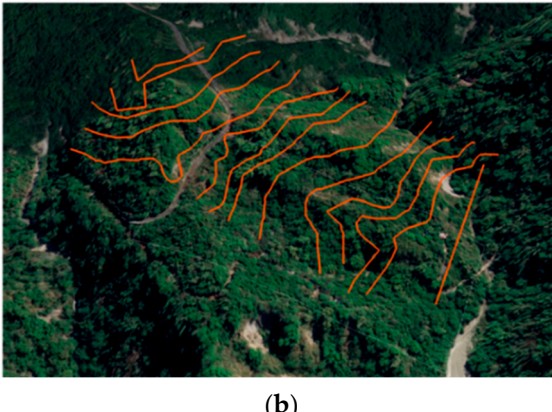

(**b**)

**Figure 4.** Case 1: Route planning results. (**a**) Grid-type mission planning, (**b**) contour-type mission planning.

The mean elevation plus 200 m was used for each route. In the contour-type, the topographic relief was judged to have exceeded the N value after the first route, so the subsequent routes were planned in the contour-type, and the topographic relief was evident from the second route to the end.

To compare and analyze the geometric differences between the contour-type and the grid-type method for the images taken in the aerial photography area, we compared the image shooting distance, overlap, and height change value in each image. The results are shown in Figure 5.

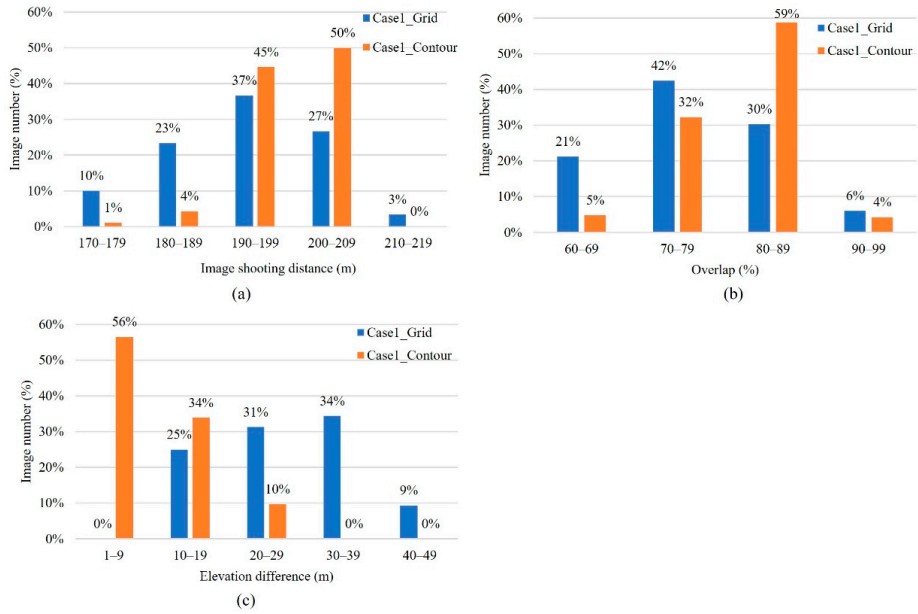

**Figure 5.** Case 1: Analysis of route planning results. (**a**) Image shooting distance, (**b**) overlap, (**c**) elevation difference.

### 3.1.1. Image Shooting Distance

The planning result of the grid-type shows that the average shooting distance is 194 m, while the average shooting distance of the contour-type is 199 m. From this result, the shooting distance of the contour-type is closer to the setting value of 200 m. The

maximum value of the grid-type is 213 m, and the minimum value is 173 m. Only 64% of the images in the 200 ± 10 m range are available. In contrast, the maximum value of 209 m and the minimum value of 178 m for the contour-type, and 95% of the images are in the 200 ± 10 m range. The results showed that the contour-type method could maintain a better shooting distance.

### 3.1.2. Overlap

The average overlap of the contour-type is 80%, while the average overlap of the grid-type is 74%. From the results in Figure 5a,b, it can be found that 70% of the images in the grid-type are below 200 m, resulting in a smaller shooting area than initially planned, which further reduces the overlap, with 63% of the images below 80% overlap. In contrast, most of the contour-type images could meet the set value, with 63% having an overlap of 80% or more.

### 3.1.3. Elevation Difference in the Same Image

The difference between the highest and lowest elevation in each image was calculated. The change was calculated in 10 m steps, as shown in Figure 5c. Figure 5c shows that 100% of the contour-type images have less than a 30 m elevation difference. In contrast, only 56% of the grid-type images have less than 30 m elevation differences. This shows that the contour-type is more effective in controlling the elevation difference in the images and can maintain the stability of aerial photography and image consistency.

### 3.2. *Case 2: Maolin District*

In Case 2, a hillside with an elevation of 1300–1550 m, a slope of 32 degrees, and an area of 130,000 m$^2$ was selected for the simulation test because there are two ravines on the hill, and the surface changes are more complicated. The grid-type and contour-type flying missions were used to simulate the route planning, and the results are shown in Figure 6.

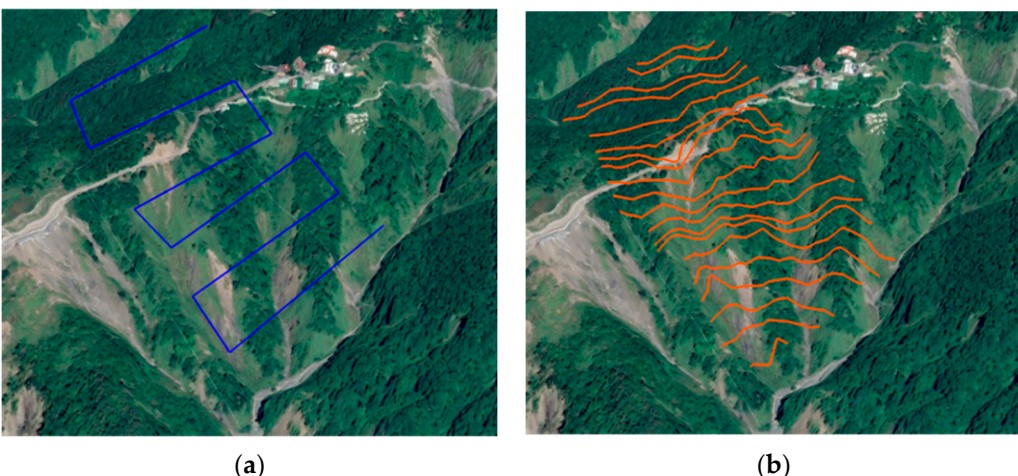

(**a**)  (**b**)

**Figure 6.** Case 2: Route planning results. (**a**) Grid-type, (**b**) contour-type.

The 6 grid-type routes with 65 images were taken, while 20 contour-type routes with 258 images were taken. The contour route maps show that the paths corresponding to the terrain changes will be adjusted with the terrain. The analysis of the results of Case 2 is shown in Figure 7.

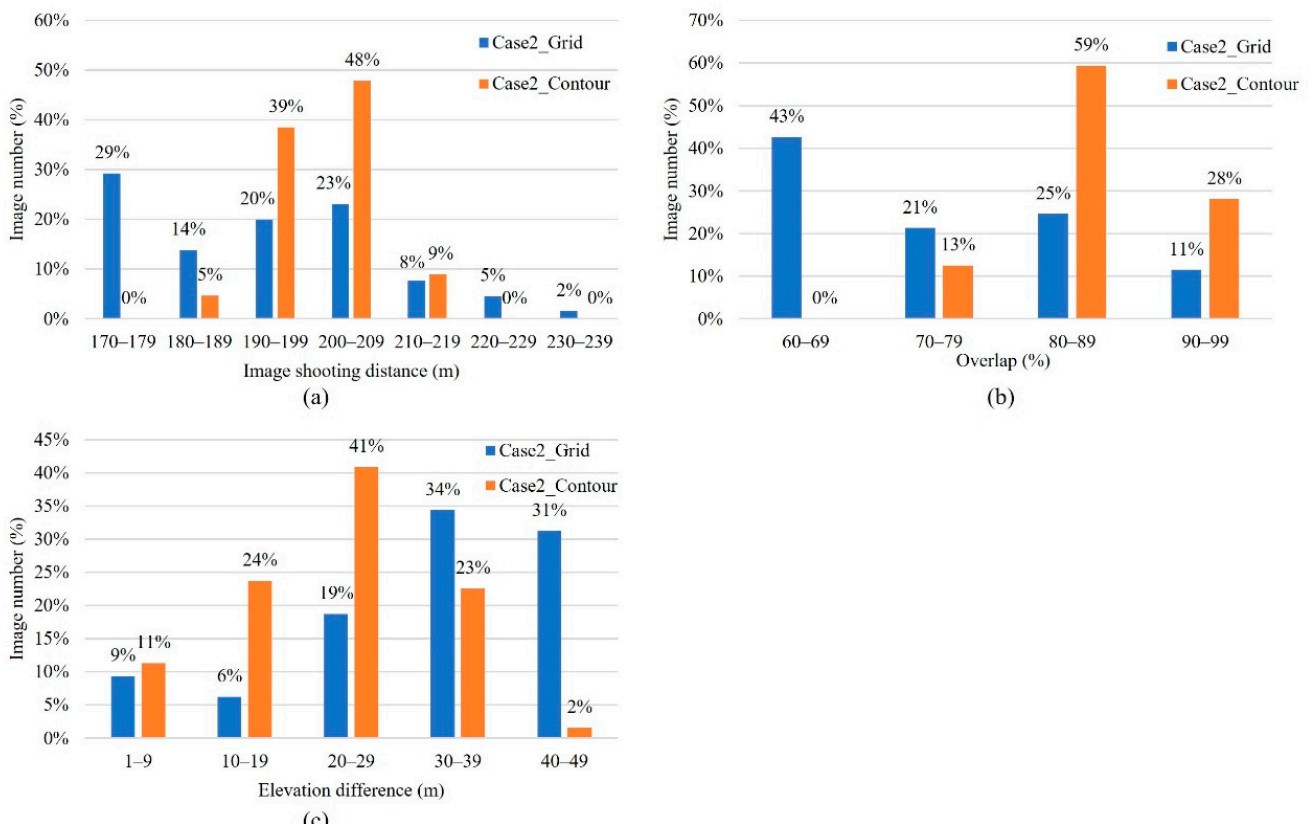

**Figure 7.** Case 2: Analysis of route planning results. (**a**) Image shooting distance, (**b**) overlap rate, (**c**) elevation difference.

### 3.2.1. Image Shooting Distance

The results of the grid-type showed a maximum distance of 234 m, a minimum distance of 113 m, and an average distance of 186 m, with 43% of the images in the 200 ± 10 m range. In addition, the contour-type results showed a maximum value of 219 m, a minimum value of 182 m, an average distance of 200 m, and images of 87% in the 200 ± 10 m range. This shows that the contour-type enables most of the images to be taken at a distance more in line with the 200 m set by the user.

### 3.2.2. Overlap

The results of the grid-type show an average overlap of 68%, with 43% of the images having a lower overlap of 60–69%. The contour-type's average overlap of 85% is to the user settings.

### 3.2.3. Elevation Difference in the Same Image

Figure 7c shows that 75% of the contour-type images have less than a 30 m elevation difference, while 35% of the grid-type images have less than a 30 m elevation difference.

The rugged sloping terrain in Case 2 is more undulating than in Case 1. The results of both terrain simulations show that the contour-type can better match the user's setting conditions and effectively reduce the effect of topography on altitude. Because of the high correlation between altitude and overlap rate, contour-type mission planning can maintain the same height and help to keep the overlap. The contour calculation process considers the amount of elevation variation, which can effectively control the elevation change within the image coverage.

## 4. Field Test

### 4.1. Study Area

To evaluate the effectiveness of grid-type and contour-type route planning methods, we chose a study site in Liugui District, Kaohsiung City, Taiwan (Figure 8). The Liugui area is characterized by undulating terrain with elevations ranging from 330 to 924 m above sea level. The upper slopes of the site exhibit a collapsed topography with well-defined erosion gullies, while the lower slopes are relatively flatter. The average slope is 38 degrees, the minimum is 14 degrees, and the maximum is 86 degrees. This terrain presents a challenging environment for UAV-based photogrammetry, as it requires careful route planning to ensure that images are captured with sufficient overlap to generate accurate terrain models. Our study assessed whether the contour-type route planning method could outperform the commonly used grid-type method regarding terrain model accuracy and completeness. By comparing the results of both methods, we aimed to identify the most suitable approach for acquiring reliable terrain information in mountainous areas.

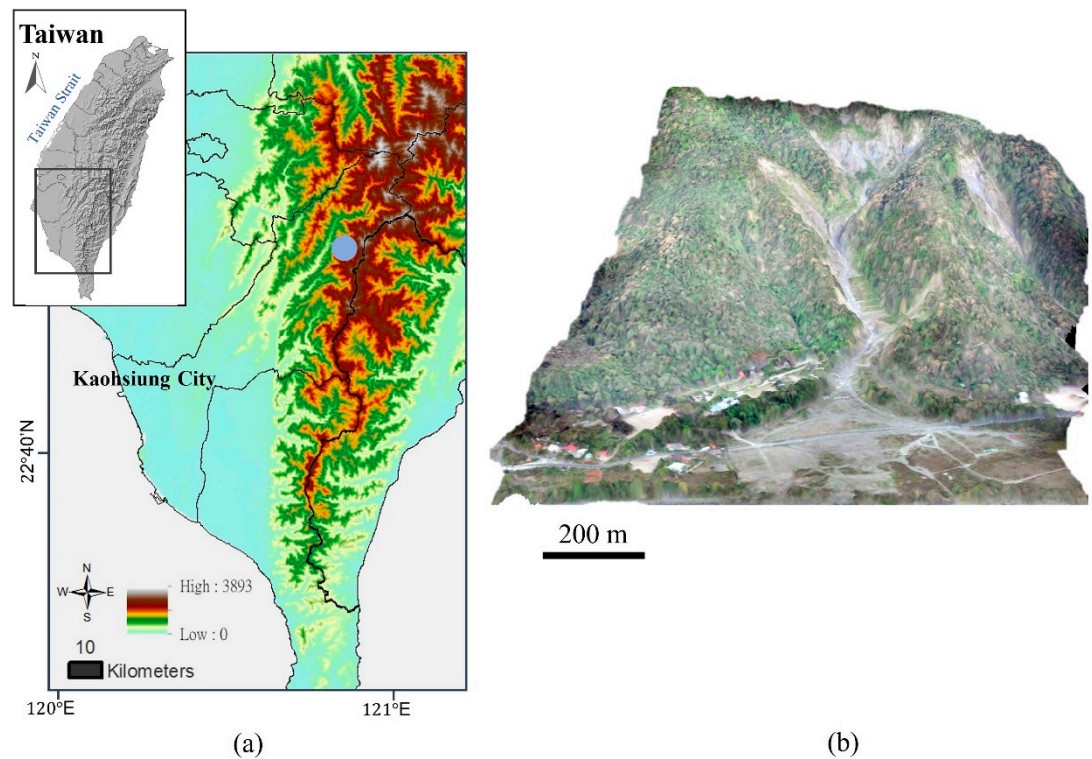

(a)  (b)

**Figure 8.** Geographical location of the study areas. (**a**) Study area location, (**b**) Liugui study area shown as a three-dimensional model.

### 4.2. Image Acquisition

Due to the contour-type flight plans, which require continuous adjustments of the drone's attitude to follow the route, precise flight control is a primary consideration to ensure compliance with the planned route. Additionally, wind conditions are typically stronger in hillside areas, and stability needs to be considered regarding the drone's length. We used a self-made 4-rotor UAV platform of 75 cm in length, equipped with an autonomous flight controller DJI N3, expanded with A3 Upgrade Kit high-performance navigation module. The flight speed was set to 4 m/s. When the UAVs operate in a mountainous area, the battery endurance will decrease due to the reduced air pressure and temperature. Sufficient flight time should be considered to complete the planned tasks for each flight to ensure flight safety. Because the Liugui area has steep gradients, the battery endurance can only be maintained for 20 min at low and 15 min at high altitudes. Thus, the flight plans were divided into three sessions. The grid-type collected 80 images during

a 45 min flight session, while the contour-type collected 166 images during a 55 min flight session. The locations of each image are shown in Figure 9.

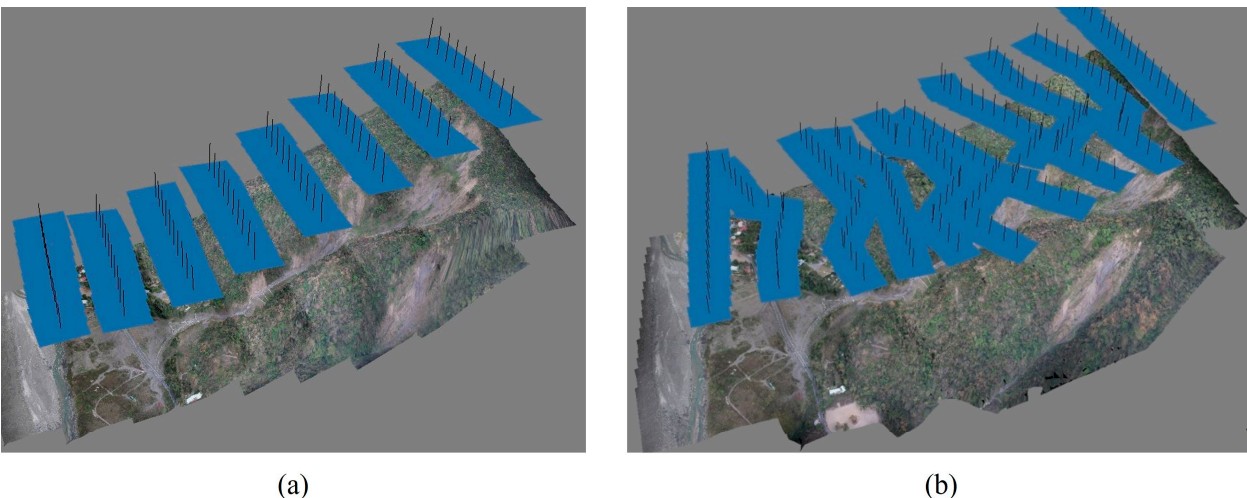

(a)          (b)

**Figure 9.** The flight route positions of the UAV images captured in the Liugui study area. (**a**) Grid-type flying mission, (**b**) contour-type flying mission.

### 4.3. Data Processing

UAVs are a commonly used technique for obtaining 3D terrain information. Through image reconstruction processing, it can generate Digital Surface Model (DSM) data from UAV images. The UAV images are generally processed using SfM software to produce a high-resolution DSM to study surface movement [9]. The SfM software calculates the intrinsic and extrinsic parameters of the images, and these values are used to calculate the positions of the points in the overlapping images to generate the 3D data. This study used the SfM software package @Agisoft Metashape [29] to produce point clouds, DSM, and orthophotos from UAV images.

This software follows a general workflow with some phases of data processing. The phases include:

(1) Importing the images into software.
(2) Alignment between overlapping images, parameter setting high accuracy, 40,000 key point limit, 4000 tie point limit.
(3) Georeferencing images using GCPs to optimize the camera position and orientation.
(4) Generation of the dense point cloud, parameter setting high quality.
(5) Building 3D meshes from point clouds.
(6) Above the 3D meshes parameter, a DSM and orthomosaic are created.

The georeferencing of the UAV images was performed using ground control points (GCPs). Since setting attributes in the forest area is complex, GCPs were placed on the bare surface with better transmittance. The GCPs were evenly distributed over a surveyed area to achieve optimal accuracy of results. We created 11 GCPs and 8 check points (CPs) distribution of the study areas, as shown in Figure 10. The GCP distribution of the study areas is shown in Figure 10. Each point used the VRS (Virtual Reference Station), and the locational accuracy was about 3 cm. In addition, the UAV images were processed by software to generate point clouds, and GCPs were added to calculate the actual location.

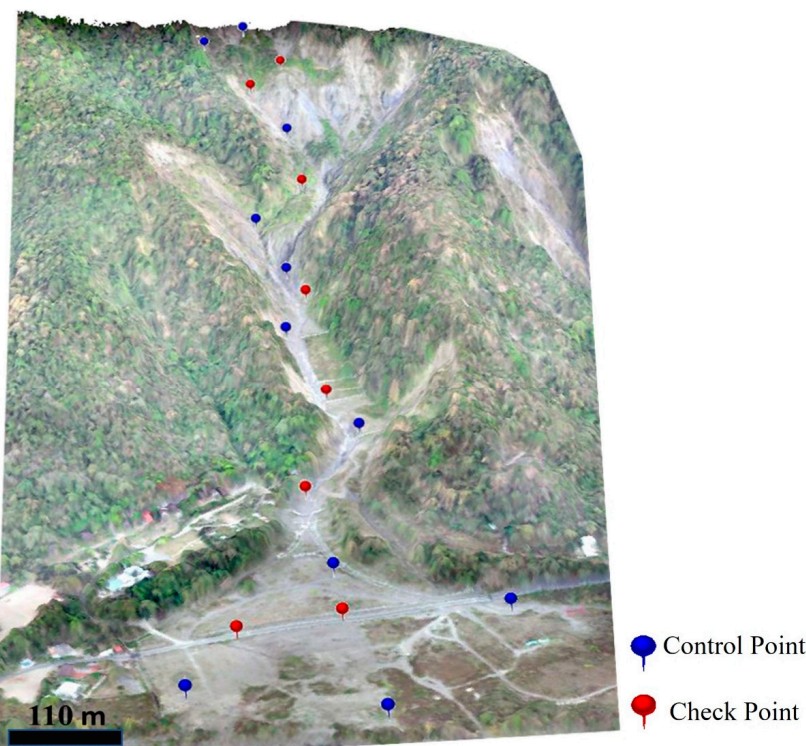

**Figure 10.** The distribution of control points and check points.

## 5. Analysis

This study aims to improve route planning to obtain high-quality UAV images. There-fore, we analyzed the captured images in terms of altitude, overlap, and elevation difference within each image and compared the results of image processing, including the size of GSD, point cloud density in particular areas, and the accuracy of the final results.

### 5.1. Flying Height and Overlay

5.1.1. Flying Height

During the bundle adjustment process, the external orientation of the UAV images is calculated using the ground control points, and the AGL altitude data are calculated by combining the elements of the exterior orientation of each image with the DSM. The altitude of each image presented in the grid-type is shown in Figure 11. The average altitude is 217 m, the highest altitude is 290 m, and the lowest altitude is 155 m, as shown in Figure 11a.

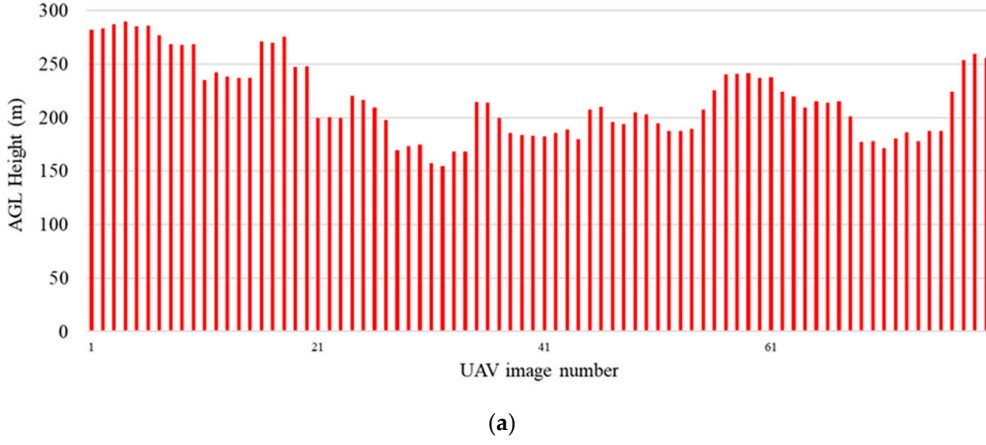

(**a**)

**Figure 11.** *Cont.*

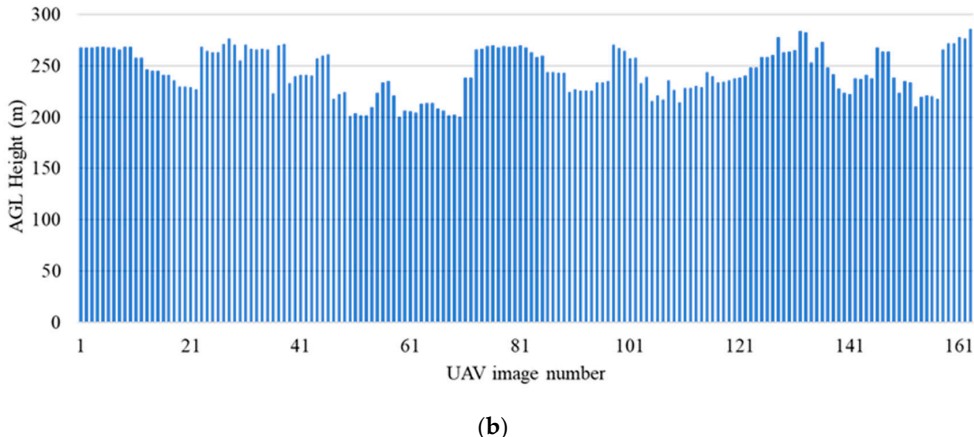

(**b**)

**Figure 11.** Flight height and the number of images. (**a**) Grid-type flying mission, (**b**) contour-type flying mission.

Using the same method of calculating the flying height as above, the results of the contour-type regularization are shown in Figure 11b, from which the average flying height is 244 m, the highest peak is 286 m, and the lowest height is 200 m.

We analyzed the height parameter; in this study, we set a height of 250 m. The results in Figure 11 show an average value of 244 m with a standard deviation of 22 m. In comparison, the grid-type method yielded an average value of 217 m with a standard deviation of 36 m. These findings suggest that the contour-type approach aligns more closely with the mission planning value, exhibiting a minor standard deviation compared to the grid-type method.

5.1.2. Overlay

The average image overlap of the grid-type is 67%, which differs from the 80% set by the mission. From the overlap results in Figure 12, we can find that the image overlap of the middle valley terrain is more significant than 80%. In comparison, the overlap of the steeper terrain on both sides is 40–60% because the valley makes the shooting away from the growth of the image coverage area more prominent so that there is a higher overlap in the high valley.

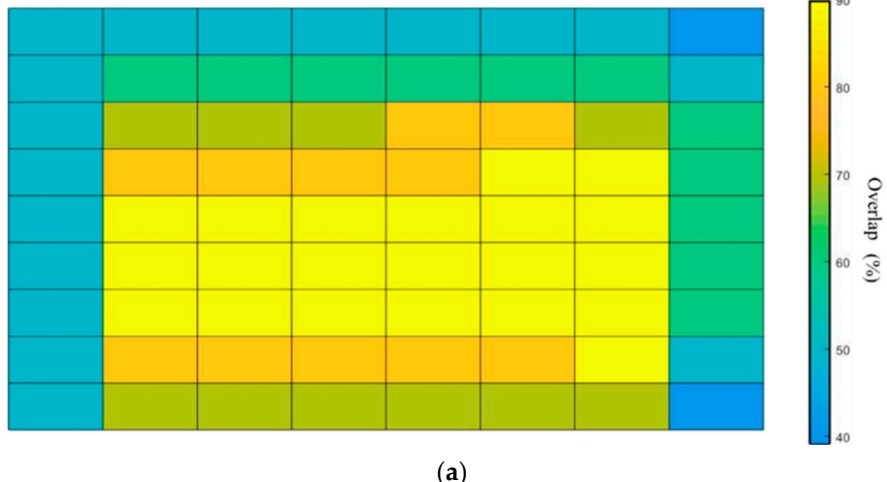

(**a**)

**Figure 12.** *Cont.*

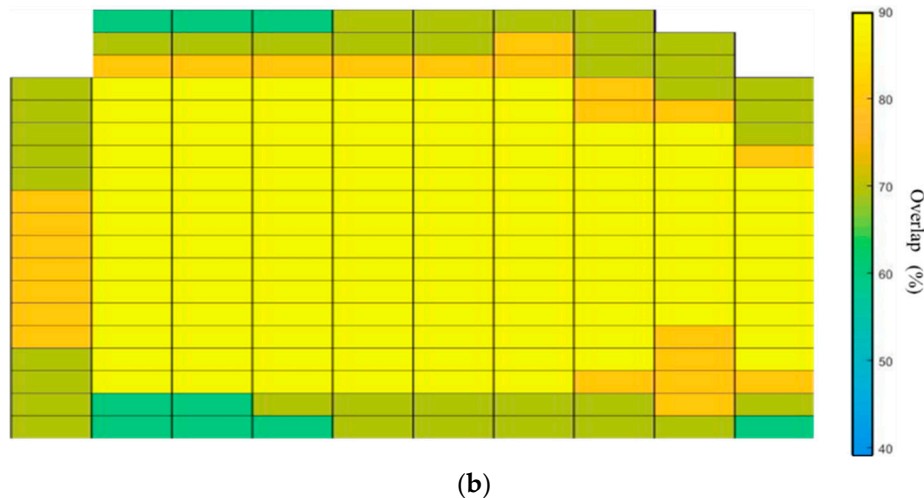

(**b**)

**Figure 12.** Results of contour route planning overlap rate. (**a**) Grid-type flying mission, (**b**) contour-type flying mission.

The results of calculating the overlap between images are shown in Figure 12, from which it can be observed that the overlap between adjacent images is similar. The data show that the average overlap is 82%, the overlap is lower only at the edge of the working area, and more than 70% overlap can be maintained in steep terrain.

### 5.2. Elevation Difference

From the location of the UAV, we can calculate the range of each image covering the ground, and the highest and lowest elevations in that range are obtained to calculate the elevation difference. The amount of elevation difference for each image is shown in Figure 13. From Figure 13a, the grid-type maximum elevation difference is 175–200 m, and 65% of the images have an elevation difference of less than 150 m. From Figure 13b, the contour-type maximum elevation difference is 125–150 m, and the elevation difference is less than 150 m for 100% of the images.

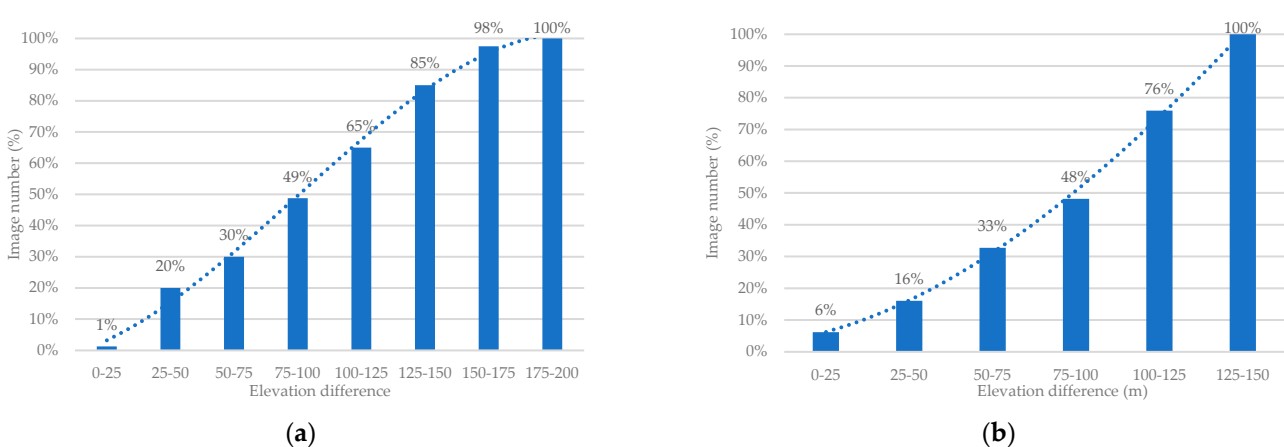

**Figure 13.** Elevation difference statistics. (**a**) Grid-type, (**b**) contour-type.

### 5.3. Efficiency Comparative

The parameters of the mission in the study are organized in Table 1. There is a significant difference in the number of captured images between the grid-type and contour-type flight plans. The contour-type captures 2–5 times more images than the grid-type, resulting in increased computational resources cost and data acquisition time. Specifically, in Case Study 1, the improvement in shooting distance and image overlap rate is not

evident. However, in Case Study 2 and the field test, the effectiveness of improving average shooting distance (14 m, 27 m) and image overlap (17%, 15%) becomes more apparent due to the presence of complex and steep terrain.

**Table 1.** Evaluating flight route planning methods.

| | Case Study 1 | | | Case Study 2 | | | Field Test | | |
|---|---|---|---|---|---|---|---|---|---|
| | **Grid** | **Contour** | **Difference** | **Grid** | **Contour** | **Difference** | **Grid** | **Contour** | **Difference** |
| Routes | 5 | 14 | 9 | 6 | 20 | 14 | 8 | 10 | 2 |
| Image number | 35 | 187 | 5 times | 65 | 258 | 4 times | 80 | 166 | 2 times |
| Average shooting distance (m) | 194 | 199 | 5 | 186 | 200 | 14 | 217 | 244 | 27 |
| Overlap (%) | 74 | 80 | 6 | 68 | 85 | 17 | 67 | 82 | 15 |
| Percentage of images with elevation difference within the threshold (%) | 56% < 30 m | 100% < 30 m | 44 | 35% < 30 m | 75% < 30 m | 40 | 65% < 150 m | 100% < 150 m | 35 |

It is noteworthy that the contour-type has a higher percentage of images with elevation differences within the threshold value within the same image compared to the grid-type. In both Case Study 1 and Case Study 2, there was a 44% and 40% increase, respectively, in the percentage of images with elevation differences less than 30 m within the same image. In the field test, 100% of the images in the contour-type were less than the threshold of 150 m of elevation difference in the same image, while only 65% of the images in the grid-type did. The contour-type increased the percentage of images by 35% over the grid-type. Although the contour-type has two times more images, the contour-type flight route planning helps to improve the elevation difference in the acquired images so that the resolution of the images can be maintained at the same quality and the accuracy of the results can be improved.

### 5.4. Ground Sampling Distance

The GSD of the image is directly affected by the elevation. Ideally, the GSD of each image should be the same, but the topographic relief will change the height and directly affect the GSD. In this study, the results of calculating the GSD from the images taken by the two methods are shown in Figure 14. The GSD of the contour-type is 4–5.7 cm, and the grid-type is 3.1–5.8 cm, as shown in Figure 14. From Figure 14, we can find that the contour-type is better than the grid-type method. Therefore, this study proposes that the contour-type method based on a DEM can effectively plan the flight height that meets the user's needs and effectively make the difference of GSD smaller.

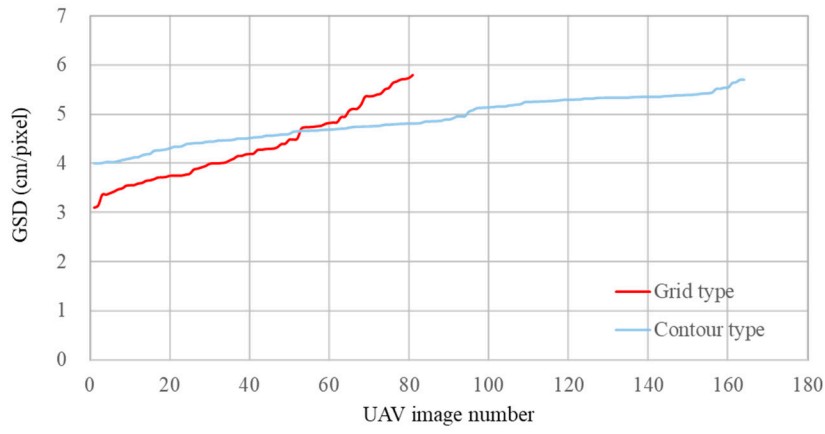

**Figure 14.** GSD statistics for two types of route planning.

In the contour route planning method, the elevation difference is considered to obtain a more consistent resolution for each captured image. From the elevation difference of each image in Figures 11–14, it can be found that the elevation difference of the contour-type is better than that of the grid-type, with a maximum gradation of 125–150 m. When the two aerial images were matched, the accuracy of contour-type matching was 1.06 pixels, and the accuracy of grid-type matching was 1.68 pixels. The minor elevation difference of the contour-type kept the resolution of the images relatively the same, which improved the accuracy of image matching.

### 5.5. Point Cloud Density and Accuracy

Two types of flight planning methods were used to produce point cloud results. The grid-type had a point density of 16 points/m$^2$, while the contour-type had a point density of 48 points/m$^2$. Further analysis of the point cloud distribution in the valley (Figure 15) showed that the grid-type had a sparse point cloud distribution with many holes. In contrast, the contour-type had a complete representation of the terrain.

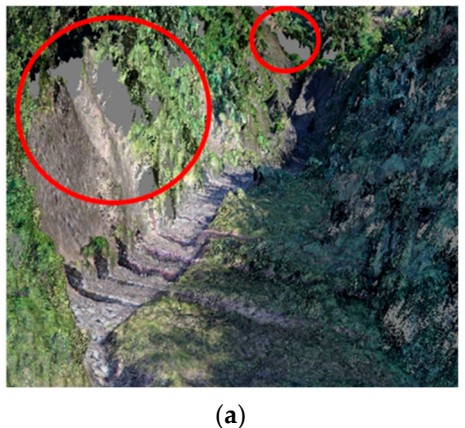 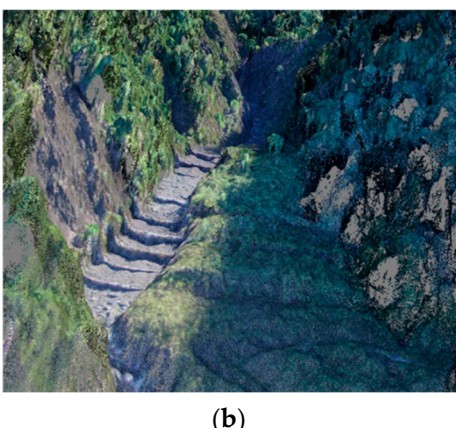

(**a**) (**b**)

**Figure 15.** The distribution of point clouds for two types of routes. (**a**) Grid-type flying mission, (**b**) contour-type flying mission. The point clouds of the grid-type are sparse, especially in the valleys (red circles) with holes in the point clouds. In contrast, the contour-type has a more complete representation.

Table 2 presents the accuracy testing results conducted after the bundle adjustment. According to Table 1, the contour-type flight plan resulted in a higher accuracy than the grid-type, with a total root mean square error (RMSE) of 11.8 cm and 16.1 cm, respectively, for the check points. This indicates that more consistent AGL height and image overlap improved reconstruction accuracy.

**Table 2.** The RMSE of control points and check points of two types of flight planning.

| | Control Points (cm) | | | Check Points (cm) | | |
|---|---|---|---|---|---|---|
| | **Horizontal** | **Elevation** | **Total RMSE** | **Horizontal** | **Elevation** | **Total RMSE** |
| **Grid** | 6.4 | 8.8 | 10.9 | 8.9 | 13.4 | 16.1 |
| **Contour** | 5.1 | 7.3 | 8.9 | 6.5 | 9.8 | 11.8 |

## 6. Conclusions

In an era of rapid development of information, how to quickly obtain three-dimensional spatial data has become the major point of research; the UAV aerial photography system, as a new aerial remote sensing tool, has unique advantages: high spatial resolution, flexibility, and mobile, high efficiency, low cost, fast map. In addition, it is also widely used for rapid, detailed investigation of land changes, disaster monitoring, and other aerial

photography operations in high-risk areas. In recent years, carrier and camera technology has constantly evolved, allowing for more convenient flight, longer aerial photography, and higher resolution cameras to quickly reconstruct UAV aerial images into a more detailed three-dimensional picture.

Nowadays, UAVs and cameras are rapidly developing and maturing, becoming new tools for acquiring geographic information. However, in the face of mountainous areas and complex terrain, if UAVs take pictures in the same traditional way, the images obtained will have different slope distances, inconsistent image proportions, and inconsistent image resolution. Therefore, excellent and effective flight planning is needed to ensure the accuracy of the subsequent 3D reconstruction. This study proposes a contour-based aerial photography method to maintain the same height as the original plan for areas with significant terrain changes.

To verify the differences between the different flight methods, we used the existing DEM to simulate the terrain and select a field with complex terrain changes for testing. The study demonstrates that the contour-based route planning method is effective in non-uniform terrain and produces an average AGL height of 244 m with a standard deviation of 22 m. The image coverage range is unaffected by uneven terrain, maintaining over 80% overlap. The reconstruction surface results show an RMSE of 11.8 cm and a point cloud density of 48 points/m$^2$. The assessed image quality indicates that the contour-based route planning method is better than the generally used grid route planning methods.

The contour method is superior to the commonly used grid method in both comparisons. Because the contour method can avoid height differences due to slope fluctuations, it can maintain a consistent flight height and has good results due to compatible image resolution, providing a good solution for mountain monitoring by UAV.

**Author Contributions:** Conceptualization, funding acquisition, methodology, writing—original draft preparation, C.-S.H., D.-Y.L. and D.-H.H.; data curation, software (data experiments), writing—review and editing, visualization, C.-S.H., D.-Y.L. and D.-H.H.; formal analysis, investigation (physical experiments), validation, C.-S.H. and D.-Y.L. All authors have read and agreed to the published version of the manuscript.

**Funding:** This research received no external funding.

**Institutional Review Board Statement:** Not applicable.

**Informed Consent Statement:** Not applicable.

**Data Availability Statement:** All data are available on request.

**Acknowledgments:** All the works were performed in NKUST. They are appreciated.

**Conflicts of Interest:** The authors declare no conflict of interest.

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
