# Peer review of "Contour Mission Flight Planning of UAV for Photogrammetric in Hillside Areas"

_applsci, doi:10.3390/app13137666_

Round 1

Reviewer 1 Report

Contour mission flight planning of UV for photogrammetric in hillside area

I suggest "hillside areas" in the title.

The paper proposes a contour mission flight plan method for UAVs based on available Digital Elevation Models (DSMs). Experiments are carried out on images collected over an undulating terrain to show the potential of the proposed method over standard flight plan approaches.

The introduction and state-of-the-art can be improved by adding a few additional references to underline the importance of the topic and clarifying which is the impact of your method and why it is necessary to present it in a scientific paper. I understand that the problem addressed in the paper can be interesting for monitoring different phenomena, not only landslides but also glaciers, in case the terrain undulation makes it difficult to acquire images with a proper GSD, overlap, etc. Lots of research have been carried out in mountain environments for monitoring landslides and glaciers with standard flight plans. Below are just a couple of examples. Are these examples of applications in which the terrain is not too steep? Is there some kind of threshold for the steepness to define when your method can really improve the results?

  • Van Tricht, L., Huybrechts, P., Van Breedam, J., Vanhulle, A., Van Oost, K., and Zekollari, H.: Estimating surface mass balance patterns from unoccupied aerial vehicle measurements in the ablation area of the Morteratsch–Pers glacier complex (Switzerland), The Cryosphere, 15, 4445–4464, https://doi.org/10.5194/tc-15-4445-2021, 2021.

  • Di Rita, M., Fugazza, D., Belloni, V., Diolaiuti, G., Scaioni, M., and Crespi, M.: GLACIER VOLUME CHANGE MONITORING FROM UAV OBSERVATIONS: ISSUES AND POTENTIALS OF STATE-OF-THE-ART TECHNIQUES, Int. Arch. Photogramm. Remote Sens. Spatial Inf. Sci., XLIII-B2-2020, 1041–1048, https://doi.org/10.5194/isprs-archives-XLIII-B2-2020-1041-2020, 2020.

  • Ioli F, Bianchi A, Cina A, De Michele C, Maschio P, Passoni D, Pinto L. Mid-Term Monitoring of Glacier’s Variations with UAVs: The Example of the Belvedere Glacier. Remote Sensing. 2022; 14(1):28. https://doi.org/10.3390/rs14010028

  • W.W. Immerzeel, P.D.A. Kraaijenbrink, J.M. Shea, A.B. Shrestha, F. Pellicciotti, M.F.P. Bierkens, S.M. de Jong, High-resolution monitoring of Himalayan glacier dynamics using unmanned aerial vehicles, Remote Sensing of Environment, Volume 150, 2014, 93-103, ISSN 0034-4257, https://doi.org/10.1016/j.rse.2014.04.025.

  • Rossi, G., Tanteri, L., Tofani, V. et al. Multitemporal UAV surveys for landslide mapping and characterization. Landslides 15, 1045–1052 (2018). https://doi.org/10.1007/s10346-018-0978-0

Page 4 line 126 GSD: Ground Sample Distance.

Page 6 line 192: Is this a repetition from the previous paragraph?

Page 6 lines 206-210: Please check the sentence.

Figures 5 and 7 in particular but this refer to all the figures: Improve quality using PDF format.

Study area: Do you have information on the steepness of the study area? This can be interesting for comparing the case study with other papers.

Image collection: Please add information on the adopted UAV and the speed of the UAV. Specify the navigation mode of the UAV (e.g. RTK, autonomous..).

Data processing: Specify information on the software settings of the processing. Also, specify the number of GCPs and CPs.

Figure 11: Clarify what are a and b in the caption.

The quality of the English is good. Minor spelling errors should be checked by the authors.

Author Response

Point 1: I suggest "hillside areas" in the title.

Response: Thanks for this suggestion. We have revised the title to "hillside areas".

--Page 1 line 3.

Point 2: The introduction and state-of-the-art can be improved by adding a few additional references to underline the importance of the topic and clarifying which is the impact of your method and why it is necessary to present it in a scientific paper. I understand that the problem addressed in the paper can be interesting for monitoring different phenomena, not only landslides but also glaciers, in case the terrain undulation makes it difficult to acquire images with a proper GSD, overlap, etc.

Response:  Thanks for this comment and suggestion. To increase the application of the article, we add the suggested literature and supplement the content description. --Page 1 lines 40, 42.

Point 3: Lots of research have been carried out in mountain environments for monitoring landslides and glaciers with standard flight plans. Below are just a couple of examples. Are these examples of applications in which the terrain is not too steep? Is there some kind of threshold for the steepness to define when your method can really improve the results?

Response:The mountainous terrain tested in this paper was retrieved from the general hillside terrain, which can provide most of the case requirements. Therefore, no particularly steep hillsides were used. In flight mission planning, many parameters affect the route design, such as flight height, camera parameters, ground resolution, etc. The information on the slope is already embedded in the value of ground horizontal and altitude. And a single slope parameter does not determine the route design, so the effect of the slope is not discussed separately in the paper. However, to let readers better understand the influence of slope, we have included the description of terrain slope in the simulation and test cases, which can let readers experience its influence. --Page 7 line 244; Page 8 line 284; Page 10 lines 323-324.

Point 4: Page 4 line 126 GSD: Ground Sample Distance.

Response:  We have revised the GSD as Ground Sample Distance. --Page 4 line 137.

Point 5: Page 6 line 192: Is this a repetition from the previous paragraph?

Response:  We have revised the description of this paragraph. [The route is a contour line when the N value is added to the route, and the corresponding elevation is higher than the D value. The contour value is the elevation of the lowest point of the previous route plus the D value.] --Page 6 lines 209-211.

Point 6: Page 6 lines 206-210: Please check the sentence.

Response: Thank you for the suggestion to revise the article to: [To verify the differential effectiveness of the contour-type flight planning method proposed in this study in mountainous areas, we will discuss it through two simulation zones. Simulation case 1 is the general slope of Nanhua District and simulation case 2 is 

the hillside terrain of Maolin District where landslides occur.] --Page 6 lines 234-238.

Point 7: Figures 5 and 7 in particular but this refer to all the figures: Improve quality using PDF format.

Response: Additional descriptions are: [The average slope is 38 degrees, the minimum slope is 14 degrees, and the maximum slope is 86 degrees.] --Page 10 lines 323-324.

Point 8: Study area: Do you have information on the steepness of the study area? This can be interesting for comparing the case study with other papers.

Response: Additional descriptions are: [The average slope is 38 degrees, the minimum slope is 14 degrees, and the maximum slope is 86 degrees.] --Page 10 lines 323-324.

Point 9: Image collection: Please add information on the adopted UAV and the speed of the UAV. Specify the navigation mode of the UAV (e.g. RTK, autonomous..).

Response: Add descriptions of the UAV device used and the speed of movement.

[We use a self-made 4-rotor UAV platform of 75 cm in length, equipped with an autonomous flight controller of DJI N3, expanded with A3 Upgrade Kit high-performance navigation module. The flight speed is set to 4 m/sec.] --Page 10 lines 339-341.

Point 10: Data processing: Specify information on the software settings of the processing. Also, specify the number of GCPs and CPs.

Response:  1. Add a description of the data processing process and settings to the article. This software follows a general workflow with some phases of data processing. Page 11 lines 360-369. 2. Also, in the article supplement the number of GCPs and CPs. --Page 12 lines 374-375.

Point 11: Figure 11: Clarify what are a and b in the caption.

Response: Added caption description of Figure 11 a and b. a: Grid-type flying mission; b: Contour-type flying mission --Page 13 line 398.

Reviewer 2 Report

The paper treats a very interesting issue and is prepared well. Some procedures need to be improved to better show the potentiality of the method. I suggest to show the data in a different way and to show the real results of the application (is in any case a good result...).

In the attached file more punctual suggestions.

Author Response

Point 1: Pag. 1, Rows 35 to 36: You used the phrase “almost in any moment”, be careful to the limitations of this technique: atmospheric condition (mist, strong wind, rain…) , day/night cycle, restricted areas.

Response 1: Thank you for your comment; we have revised the sentence.

Page 1 lines 35-37.

Point 2: Pag. 1, Rows 38 to 39: The phrase “the literature shows” need more than only one reference.

Response 2: Thank you for your comment; we have integrated the description of this reference with the previous sentence and expanded the applied reference data.

Page1 lines 38-40.

Point 3: Pag. 2, Row 82: An introduction paragraph to chapter 2 "methodology" can be useful to briefly summarize the methodologies you have used.

Response 3: Thank you for your comment; an introduction of the content has been added.

Page 2 lines 89-92.

Point 4: Pag. 3, Row 90: In Fig.1 you called “h” every height, did you mean “h1”, “h2”, “h3”. In the same figure the label “slope” is not clear to what is referred.

Response 4: Thank you for suggesting redraw the schematic of Fig. 1 and make the symbol number consistent with the route number.

Page 3 line 101.

Point 5: Pag. 2, Row 102: In Fig.2 the transverse flight to the slope is schematized (the course equivalent of Rn of Fig.1), you should specify it to better correlate the figures.

Response 5: We modified the symbols in Fig. 2 to be the same as in Fig. 1 and explained their correlation.

Page 3 lines 113-115.

Point 6. Pag. 3, Row 109: From your description and the flowchart in Fig.3 is not clear what specifically does the method you propose make use of (software, algorithms, other...)?

Response 6:  Thank you for suggesting adding the software and data needed for this study.

[This study utilized the ASTER DEM as the terrain reference data, with the coordinate system set to WGS-84. The DEM was extracted based on the user-defined flight area. The flight route was then generated using the defined camera parameters, ground sample distance (GSD), and overlap rate, following the workflow outlined in Figure S1-S7. The calculations were performed using Python code, resulting in a KML file format. The IPad’s DJI Ground Station Pro app was used to import the KML file and create the flight mission, allowing flight speed selection. Subsequently, the UAV executed the flight mission to capture aerial images.]

Page 6 lines 226-233.

Point 7:  Pag. 3, Row 110: The phrase “the steps shown in flowcharts S1-S6” should be converted in: “the steps S1-S6 shown in flowchart”. Check if the meaning is what you mean.

Response 7:  We have revised it.

Page 4 line 120.

Point 8: Pag. 4, Row 121: The flowchart (Fig.3) at S4 has only two allowed options (Yes = go to S5, No = go to S6):

(a) how do you get to the "Successful" step,

(b) how do you proceed once you get to S5/S6 if there are no outgoing arrows?

Response 8: Thank you for the comment. Redrawing of the flowchart (Fig. 3) with the addition of the S7 step to determine the conditions for finishing the calculation. Page 4 line 131.

A new description has been added.

Page 6 lines 213-215.

Point 9: Pag. 4, Rows 134 to 136: Regarding the percentages of overlays suggested by Agisoft Metashape and Pix4D: indicate the sources and insert the corresponding references.

Response 9: Supplements the Agisoft Metashape and Pix4D references as [30] and [31].

Page 5 lines 145 and 146.

Point 10:  Pag. 5, Row 142: SRTM and ASTER: bibliographic references are missing. You could also include a short explanation.

Response 10: Thank you for the suggestion to add full names and references.

SRTM: Shuttle Radar Topographic Mission.

Page 5 line 153.

ASTER: Advanced Spaceborne Thermal Emission and Reflection 

Radiometer.

Page 5 line 154.

Point 11: Pag. 5, Row 152: I think a “.” was intended respect to “°”.

Response 11:  Thank you for your comment. It has been revised.

Page 5 line 168.

Point 12:  Pag. 6, Rows 192 to 195: The paragraph “6. S6 Contour route” has the same text of the previous paragraph “5. S5 Straight route”.

Response 12: The repetitive paragraph has been corrected.

[The route is a contour line when the N value is added to the route, and the corresponding elevation is higher than the D value. The contour value is the elevation of the lowest point of the previous route plus the D value.]

Page 6 lines 209-211.

Point 13:  Pag. 6, Row 216 and Pag. 8, Row 256: If you use the system in meters, put the area in m^2 for consistency.

Response 13: Revised to m2.

Page 7 line 245 and Page 8 line 285.

Point 14: Pag. 6, Rows 217 to 218: Change “five” into “5” for consistency with the “14”.

Response 14 : Revised to make consistency.

Page 6 line 246.

Point 15: Pag. 7, Row 228: The graphs in Fig.5 are nor readable (blurry).

Response 15: Figures 5 were adjusted to improve their clarity.

Page 7 line 258.

Point 16:  Pag. 15, Row 410 to 411: The phrase "the focus of research" means very exclusively, a revisiting of the phrase is recommended.

Response 16: Thank you for your comments, and revise the phrase to "major point of research".

Page 16 lines 482-483.

Point 17:  Pag. 15, Rows 430 to 431: In all the previous text of the article you have never talked about "standard deviation", if you cite data in the conclusions they must first be presented.

Response 17 : Thank you for the suggestion to add standard deviation information to the analysis.

Page 13 line 400-405.

Point 18: In bibliography are present reference named 26 and 27 but they are not cited in the text.

Response 18: The citation has been modified.

The reference number 26 has been changed to 31, and the reference text on Page 5 line 155.

The reference number 26 has been changed to 33, and the reference text on Page 6 line 196.

Point 19: 

QUESTION 1 

Have you considered the technological challenge for commercial drones in maintaining complex routes (as for contour type) compared to linear routes (in the grid type)? Which hardware did you use for the test, and which are the technical specifications? 

Response 19: Because the focus of commercial drones is on most people's needs, it is not easy to care for the few users who require high precision. The method proposed in this paper will increase the problems of the flight path, time, and direction adjustment, and the hardware requirements are higher than usual. However, due to the improvement of the current technology, as long as the conditions can be met, it is possible to perform the related tasks. The hardware equipment used in this study has been added to the paper.

Page 10 lines 339-341.

Point 20:  QUESTION 2 

1From what is written (Pag 6 Rows 216 to 218, Pag 8 Rows 261-264, Pag 10 Rows 310-312) (and summarized in tab. as follow:

  case study 1 case study 2 field test
  grid         contour grid       contour grid     contour
route 5                 14 6               20  
image number 35                187 65             258 80           166

The grid-type and contour-type flight planes differ significantly in the number of images taken, this is evident both in the simulation (case study 1 and case study 2) and in the field test. To relate the data (Fig. 5 and Fig. 7) regarding "grid type" and "contour type" you have normalized and used a percentage scale. The same goes for the evaluation of the field test (Fig 12, Fig 14). These differences should be discussed and considered in Chapters 5 (analysis) and 6 (conclusions) to avoid incurring an overestimation of the parameters in favor of the contour-type compared to the grid-type.

Response 20: Thank you for your suggestions. In addition to the original comparison, Table 1 was added to illustrate its advantages, and the benefit analysis in Section 5.3 was added to highlight the method.

Page 15 lines 428-446.

Reviewer 3 Report

1. In my opinion the authors should improve literature review to compare/introduce their method in the context of other authors’ works.

2. Fig. 3. In my opinion, the ‘Successful’ diagram block should not be connected with the decision block in such way. Loop is not properly shown on the diagram as well.

3. Please write what might be the origin of a DEM model. What is the source of it. How it can be obtained by your method users.

4. Please explain abbreviations: SRTM, ASTER, RMSE.

5. In my opinion, formatting of punctuations in section 2.2 might be improved.

6. Please describe equations in a uniform way.

7. Page 6, lines 186-195 – there is no difference in description of S5 and S6.

8. Quality of figures 5 and 7 is bad – labels are barely/not readable.

9. Please put spaces between measurement value and unit.

10. Why simulations and field tests do not consider the same places (slopes)? It needs some better explanation.

11. Lines 317-324 – please improve style.

12. Line 320 – improper citation style.

13. References 26 and 27 are not used in the text.

14. In my opinion, the authors should use statistical tests, while analysing the data, to tell which method is better. Averages are not the best way of comparing things.

Minor editing of English language required

Author Response

Point 1: In my opinion the authors should improve literature review to compare/introduce their method in the context of other authors’ works.

Response 1: Thank you for your comment. We have rewritten the literature review and introduced their route planning in the hillside area.

Page 2 line 59-81.

Point 2: Fig. 3. In my opinion, the ‘Successful’ diagram block should not be connected with the decision block in such way. Loop is not properly shown on the diagram as well.

Response 2: Thanks to your comment, we have redrawn the flowchart in Fig. 3, adding the S7 step to determine the condition for successfully completing the calculation.

Page 4 line 131.

Point 3: Please write what might be the origin of a DEM model. What is the source of it. How it can be obtained by your method users.

Response 3 : The main DEM sources are ASTER and SRTM, described on Page 5, lines 151-159.

"The ASTER (1 arc second) DEM was selected for the study. ASTER DEM data is to access publicly available data repositories such as the NASA Earth Observing System Data. Users can download the desired DEM files from the platform in the appropriate flight area."

Point 4: Please explain abbreviations: SRTM, ASTER, RMSE.

Response 4: Thank you for your comment. We have revised the abbreviations.

SRTM: Shuttle Radar Topographic Mission (Page 5 line 153)

ASTER: Advanced Spaceborne Thermal Emission and Reflection Radiometer (Page 5 line 154)

RMSE: Root mean square error(Page 16 line 476)

Point 5: In my opinion, formatting of punctuations in section 2.2 might be improved.

Response 5: Thank you for suggesting that we standardize the format of the descriptions.

Pages 3-6 lines 116-233.

Point 6 : Please describe equations in a uniform way.

Response 6: Thank you for suggesting that we align the parameters and format of the equations.

Page 5 lines 165,173,193, and Page 6 line 197.

Point 7: Page 6, lines 186-195 – there is no difference in description of S5 and S6.

Response 7: We have revised the description of S6.

"The route is a contour line when the N value is added to the route, and the corresponding elevation is higher than the D value. The contour value is the elevation of the lowest point of the previous route plus the D value."

Page 6 lines 209-211.

Point 8: Quality of figures 5 and 7 is bad – labels are barely/not readable.

Response 8: Figures 5 and 7 were adjusted to improve their quality.

Page 7 line 258 and Page 9 line 293.

Point 9: Please put spaces between measurement value and unit.

Response 9: Thank you for your suggestion; we have revised the format.

Point 10: Why simulations and field tests do not consider the same places (slopes)? It needs some better explanation.

Response 10: Thank you for this comment; added the description as

"To verify the differential effectiveness of the contour-type flight planning method proposed in this study in mountainous areas, we will discuss it through two simulation zones. Simulation case 1 is the general slope of Nanhua District and simulation case 2 is the hillside terrain of Maolin District where landslides occur. Two regions with different surface changes were selected."

Page 6 lines 235-239.

The field test site is selected from the Liugui District, which has a significant surface change. To increase the reader's understanding of the area's topography, the slope is added.

Page 10 lines 323-324.

Point 11: Lines 317-324 – please improve style.

Response 11: We have revised the description.

Page 11 lines 352-359.

Point 12: Line 320 – improper citation style.

Response 12: We have revised it.

Page 11 line 355.

Point 13: References 26 and 27 are not used in the text.

Response 13: The citation has been modified.

The reference number 26 has been changed to 31, and the reference text on Page 5 line 155.

The reference number 26 has been changed to 33, and the reference text on Page 6 line 196.

Point 14: In my opinion, the authors should use statistical tests, while analysing the data, to tell which method is better. Averages are not the best way of comparing things.

Response 14 : Thank you for your suggestions. In addition to the original comparison, the standard deviation value is increased. (Page 13 line 400-405.)

And Table 1 was added to illustrate its advantages, and the benefit analysis in Section 5.3 was added to highlight the method. Page 15 line 445.

Round 2

Reviewer 1 Report

The figure quality should be improved. I suggest the use of PDF format for all the figures, especially Figure 3. Figure 13 is not well visible

Author Response

Point 1: The figure quality should be improved. I suggest the use of PDF format for all the figures, especially Figure 3. Figure 13 is not well visible.

Response

Thank you for your suggestion. After testing, we found that using the original image and adjusting the size worked better, so we updated the figure file this way.

Fig. 1-2, Page 3 lines 101,.113; Fig. 3, Page 4 line 131; Fig. 4, Page 7 line 247; Fig. 5, Page 8 line 260; Fig. 6-7, Page 9 lines 290, 295; Fig.8-9, Page 11, lines 334, 351; Fig. 10, Page 12 line 382; Fig. 11, Page 13 line 401; Fig. 12, Page 14 line 420; Fig. 13, Page 15 line 431; Fig. 14-15, Page 16 lines 462;478.

Reviewer 3 Report

Lines 162-175 - if I understand correctly, these lines are the comment for "Flight height". If yes, then indentation is wrong.

In my opinion Fig. 5 should not be split between pages.

Line 343 - the authors wrote, that "When the UAVs operate in a mountainous area, the battery endurance will decrease due to the increased air density." How it compares to encyclopedic knowledge being "Air density, like air pressure, decreases with increasing altitude."?

Table 1 and comment to it (Field test - Difference) - why suddenly 100-65 equals 40? In the previous cases math was ruthlessly accurate. For me, interpretation of Table 1 should be improved. It is not enough clear right now. For example: "In the field test, the height differences are less than 150 m, resulting in a similar 40% increase in images" - it is not so clear that the images meet criterium being height differences are less than 150 m on one image. Whatmore from the data it looks like 100% of images meet the criterium of height differences being less than 150 m on one image (contour method), witch is 35% better than in the case of the 2nd method (grid method), but the recommendation is to take bigger number of images aimed at reducing elevation differences in the case of the better method.

Author Response

Point 1: Lines 162-175 - if I understand correctly, these lines are the comment for "Flight height". If yes, then indentation is wrong. 

Response

Thank you for your comment and for shortening the format of the narrative to the correct position.

Page 5 lines 166-174.

Point 2: In my opinion Fig. 5 should not be split between pages.

Response

Thanks for the comment to adjust Fig.5 and the captions to the same page.

Page 8 line 260.

Point 3: Line 343 - the authors wrote, that "When the UAVs operate in a mountainous area, the battery endurance will decrease due to the increased air density." How it compares to encyclopedic knowledge being "Air density, like air pressure, decreases with increasing altitude."?

Response

Thank you for your comment. During aerial photography in mountainous areas, factors such as lower air pressure and lower temperature will affect the operating efficiency and time of the battery. Therefore, revise the text to

"When the UAVs operate in a mountainous area, the battery endurance will decrease due to the reduced air pressure and temperature."

Page 11 lines 344-345.

Point 4: Table 1 and comment to it (Field test - Difference) - why suddenly 100-65 equals 40? In the previous cases math was ruthlessly accurate. For me, interpretation of Table 1 should be improved. It is not enough clear right now. For example: "In the field test, the height differences are less than 150 m, resulting in a similar 40% increase in images" - it is not so clear that the images meet criterium being height differences are less than 150 m on one image. Whatmore from the data it looks like 100% of images meet the criterium of height differences being less than 150 m on one image (contour method), witch is 35% better than in the case of the 2nd method (grid method), but the recommendation is to take bigger number of images aimed at reducing elevation differences in the case of the better method.

Response

  1. Thank you for your comment. The difference value has been revised to 35%.

Page 15 line 452.

  1. Thank you for your comment. In order to clearly explain the contents of Table 1, we revised the Table 1 field name and text description. Field name as "Percentage of images with elevation difference within the threshold (%)”.

Page 15 line 452.

  1. And text description revised as

"It is noteworthy that the contour-type has a higher percentage of images with elevation differences within the threshold value within the same image compared to the grid-type. In both Case Study 1 and Case Study 2, there was a 44% and 40% increase in the percentage of images with elevation differences less than 30 m within the same image.  In the field test, 100% of the images in the contour-type were less than the threshold of 150 m of elevation difference in the same image, while only 65% of the images in the grid-type did.  The contour-type increased the percentage of images by 35% over the grid-type. Although the contour-type will have 2 times more images, the contour-type flight route planning helps to improve the elevation difference in the acquired images so that the resolution of the images can be maintained at the same quality and the accuracy of the results can be improved."

Page 15 lines 441-451.
